# Aerosol Delivery of Palivizumab in a Neonatal Lamb Model of Respiratory Syncytial Virus Infection

**DOI:** 10.3390/v15112276

**Published:** 2023-11-19

**Authors:** Hasindu S. Edirisinghe, Anushi E. Rajapaksa, Simon G. Royce, Magdy Sourial, Robert J. Bischof, Jeremy Anderson, Gulcan Sarila, Cattram D. Nguyen, Kim Mulholland, Lien Anh Ha Do, Paul V. Licciardi

**Affiliations:** 1Murdoch Children’s Research Institute, Melbourne 3052, Australia; hasindu.edirisinghe@mcri.edu.au (H.S.E.); magdy.sourial@mcri.edu.au (M.S.); jeremy.anderson@mcri.edu.au (J.A.); gulcan.sarila@mcri.edu.au (G.S.); cattram.nguyen@mcri.edu.au (C.D.N.); kim.mulholland@lshtm.ac.uk (K.M.); lienanhha.do@mcri.edu.au (L.A.H.D.); 2Department of Paediatrics, University of Melbourne, Melbourne 3010, Australia; 3Royal Children’s Hospital, Melbourne 3052, Australia; 4Royal Women’s Hospital, Melbourne 3052, Australia; 5Monash Biomedicine Discovery Institute, Monash University, Melbourne 3168, Australia; simon.royce@monash.edu; 6Institute of Innovation, Science and Sustainability, Federation University, Melbourne 3806, Australia; r.bischof@federation.edu.au

**Keywords:** respiratory syncytial virus, lamb model, lung, aerosolisation, palivizumab

## Abstract

(1) Background: Palivizumab has been an approved preventative monoclonal antibody for respiratory syncytial virus (RSV) infection for over two decades. However, due to its high cost and requirement for multiple intramuscular injections, its use has been limited mostly to high-income countries. Following our previous study showing the successful lung deposition of aerosolised palivizumab in lambs, this current study evaluated the “proof-of-principle” effect of aerosolised palivizumab delivered as a therapeutic to neonatal lambs following RSV infection. (2) Methods: Neonatal lambs were intranasally inoculated with RSV-A2 on day 0 (day 3 post-birth) and treated with aerosolised palivizumab 3 days later (day 3 post-inoculation). Clinical symptoms, RSV viral load and inflammatory response were measured post-inoculation. (3) Results: Aerosolised therapeutic delivery of palivizumab did not reduce RSV viral loads in the nasopharynx nor the bronchoalveolar lavage fluid, but resulted in a modest reduction in inflammatory response at day 6 post-inoculation compared with untreated lambs. (4) Conclusions: This proof-of-principle study shows some evidence of aerosolised palivizumab reducing RSV inflammation, but further studies using optimized protocols are needed in order to validate these findings.

## 1. Introduction

Respiratory syncytial virus (RSV) is the leading cause of lower respiratory tract infections (LRTIs) in children under five years of age [1,2,3]. RSV causes ~6.7% of deaths annually in infants younger than 1 year old worldwide, with 99% of these deaths occurring in low- and middle-income countries (LMICs) [4,5,6].

Currently, there are no approved RSV treatments for infants. The two recently approved RSV preventatives, a long-acting monoclonal antibody (Niservimab, Sanofi, Astra Zeneca, Cambridge, UK) for newborns and a maternal vaccine (RSVPreF, Pfizer, NY, USA), have high costs which are out of the reach of many LMICs. The risk of preterm births associated with RSVPreF remains a concern and requires more data, but is currently approved for use in pregnant women at 32–36 weeks of gestation [7,8,9,10]. Given their high costs, these two newly approved preventions are not likely to be available in LMICs. Palivizumab (Synagis^®^, Sobi, MA, USA) is also a monoclonal antibody which has been used for more than 20 years for RSV prophylaxis, mainly in high-income countries, but is restricted to only very high-risk infants. The requirement of repeated monthly intramuscular injections (IM) before each RSV season and high cost have been critical limitations to its broader use among infants [11].

Despite these newer RSV preventives, there remains a need to identify alternative cost-effective strategies to prevent RSV infection and severe disease in infants. Aerosol delivery represents one such approach. We previously demonstrated the deposition of aerosolised palivizumab into the lungs of lambs, suggesting that this might be a feasible approach by which to combat respiratory infections while potentially reducing the dose and cost of palivizumab using a non-invasive delivery route [12]. Due to the significant similarities to humans in lung structure and function, neonatal lambs serve as a useful large animal model in which to study RSV infection [13,14,15,16,17].

This study aimed to evaluate the effect of aerosolised palivizumab when given as a treatment in a newborn lamb model of RSV infection. We hypothesised that the delivery of therapeutic aerosolised palivizumab would reduce RSV viral load and lung inflammation.

## 2. Materials and Methods

### 2.1. Study Design

This study was approved by the Murdoch Children’s Research Institute (MCRI) Animal Ethics Committee (A895). The experiments were designed and are reported with reference to the ARRIVE (Animal Research: Reporting of in vivo Experiments) guidelines [18].

An overview of the RSV inoculation, treatment, and sampling regime for this experiment is summarised in Figure 1. A total of 25 newborn lambs were allocated to five groups (Table 1). Group 0 lambs served as uninfected, untreated controls. For intranasal RSV inoculation (Group 1 and Group 2), the human RSV-A2 strain (obtained from ATCC, VA, USA) [19] was prepared in sterile saline and delivered by gentle infusion via a syringe into each nostril (maximum volume of 2 mL/nostril, total of 102 × 10^6^ PFU inoculation per lamb) on day 0 (day 3 post-birth). Group 1 lambs served as RSV-infected, untreated controls (RSV/untreated). Group 2 lambs were infected with RSV and treated with aerosolised palivizumab (RSV/palivizumab) (Synagis^®^, Sobi, MA, USA) while under sedation, which was performed on day 3 post-inoculation using an Aeroneb Go^®^ (Aerogen, Galway, Ireland) nebuliser (15 mg/kg diluted to 10 mg/mL in sterile saline) based on our previous study [12].

### 2.2. In-Life Measures and Sample Collection for Assessment of RSV Infection

Clinical parameters were measured using a calibrated weighing scale (body weight), digital rectal thermometer (body temperature), and pulse oximetry for oxygen saturation. Nasopharyngeal (NP) swabs were collected between day 0 to day 10 post-inoculation from the posterior nasal cavity of both nostrils using a sterile neonatal flocked swab, then stored in RNALater at −80 °C. Lambs were anaesthetised with ketamine, which was administered intramuscularly, for the collection of bronchoalveolar lavage fluid (BALF) samples on days 0, 2, 6, and 10 post-inoculation. Autopsy tissue samples were collected either at day 6 (peak infection) or day 10 (resolution of infection) from all lambs. Tissues from the cranial, middle, caudal, and accessory lung lobes were fixed in 4% (*w/v*) paraformaldehyde for histopathological analysis.

### 2.3. RSV RT-qPCR

Viral RNA was extracted from NP swabs and BALF samples using a QIAamp Viral RNA Mini Kit (Qiagen, Hilden, Germany), then processed for quantitative RT-PCR using a published protocol [20].

### 2.4. Lung Histopathology

The slides were deparaffinized, stained with haematoxylin and eosin (H&E), and cover-slipped with Depex^TM^. Additional slides were stained with Alcian blue and periodic acid schiff (AB-PAS).

### 2.5. Light Microscopy Analysis

The stained slides were scanned at 40× magnification on ScanScope (ScanScope XT, Leica Aperio Technologies) and analysed using Imagescope v.12.1 (Leica Aperio Technologies, Nussloch, Germany) software. A histological score was given to all lambs according to the inflammation and congestion observed after H&E staining. A representative image was considered for each score to assess three fields of view of the lung regions independently. An average inflammation/congestion score was then calculated per lamb in each group to compare the effect of palivizumab using the scoring matrix below, which was adapted from [21]. Validation of the analysis was carried out by an independent operator in a blinded manner.

Overall consolidation.

0—None;

1—1–10% consolidation;

2—10–25% consolidation;

3—25–50% consolidation;

4—50–75% consolidation;

5—>75% consolidation.

AB-PAS stain was used for the identification of goblet cells (GC) in the lung tissue. Five cartilaginous and non-cartilaginous airways in each lung section were used for manual GC counts under 40× magnification. A mean number of cells per 100 μm basement membrane length per lamb in each group was then calculated.

### 2.6. Statistical Analysis

Clinical data (body temperature, body weight, oxygen saturation) were presented as the mean ± SEM. Viral load and histopathology data were presented as the median ± interquartile range (IQR), as appropriate. Differences between groups were analysed using the Mann–Whitney test. A *p* value < 0.05 was considered statistically significant. All of the results are graphically presented with GraphPad Prism version 7 (GraphPad Software, San Diego, CA, USA).

## 3. Results

### 3.1. Aerosol Administration

Lambs in the RSV/palivizumab group (Group 2) received aerosolised palivizumab on day 3 post-inoculation via an Aeroneb Go^®^ device (Figure 2A). The mean dose (±SD) administered to the lambs was 113 (±16) mg/kg, with an estimated mean nebulised mass (±SD) of 79 (±17) mg/mL over 27 (±5) min.

### 3.2. Clinical Observations

No clinical signs of RSV infection were observed in any of the inoculated lambs up to day 6 post-inoculation (Figure 2B–D). The increases in body weight were within the healthy range of 240 g–280 g per day for all lambs, with no difference between the groups. Body temperature was within the normal range of 38–40 °C, with a recorded average baseline temperature of 39 °C across all groups. Oxygen saturation, as measured by pulse oximetry, was within the normal range of 95–100% at all times, with clinically irrelevant fluctuations in lambs across all groups.

### 3.3. Viral Load

RSV was detected in RSV-inoculated lambs only (groups 1 and 2) in NP swabs and BALF samples (Figure 3). On day 6 post-inoculation (stable peak infection), RSV/palivizumab lambs (Group 2) had an 85.2% lower median viral load of 3.1 × 10^2^ copies RNA/mL compared to the RSV/untreated lambs (Group 1), which had a median viral load of 2.1 × 10^3^ copies RNA/mL, although this was not significant (Figure 3A).

The BALF viral load values were also evaluated for each group (Figure 3B). On day 6, the median RSV load was similar across all groups. There were three lambs that showed detectable viral loads in the RSV/untreated group (group 1) and two in the RSV/palivizumab group (group 2).

Viral load kinetics according to NP swabs and BALF over the 10-day post-inoculation period are shown in the Appendix A.

### 3.4. Lung Histopathology

Inflammation scores were assessed for each group (Figure 4). In Figure 4A–C, a representative image of lung inflammation is shown for the three groups. On day 6, additional alveolar spaces and thin alveolar walls were seen in the control (group 0) lambs (median ± IQR of 1.3 ± 0.2), whereas the RSV/untreated (group 1a) lambs showed significantly increased (*p* = 0.035) congestion and lung pathologies, with thick alveolar walls and reduced alveolar spaces (median ± IQR of 3.1 ± 1.4). For the RSV/palivizumab (group 2a) lambs, the inflammation scores were significantly lower than those of the RSV/untreated lambs (median ± IQR of 1.7 ± 0.8; *p* = 0.041) (Figure 4D).

AB-PAS stains were utilized to identify goblet cell (GC) mucin as a marker of RSV histopathology on day 6 post-inoculation (Figure 5). In Figure 5A–C, a representative image of cartilaginous and non-cartilaginous airways across the three groups is shown. There was a non-significant increase in GC observed in the cartilaginous airways of the RSV/untreated (Group 1a) lambs (median ± IQR 2.6 ± 0.8) compared to the RSV/palivizumab lambs in Group 2a (median ± IQR 1.9 ± 0.8; Figure 5D). The average GC counts per 100 μm in the non-cartilaginous airways were similar across all groups.

## 4. Discussion

This study examined the effect of therapeutic aerosolised palivizumab in a lamb model of RSV infection. Lambs were infected with high-dose administration of RSV to the upper airways. No clinical symptoms were observed in these lambs following RSV inoculation. On day 6 post-inoculation (i.e., at peak infection), there was no difference in viral loads between the untreated group and the group treated with aerosolised palivizumab. However, significantly reduced inflammation scores in the lungs were observed for the palivizumab-treated group compared to the untreated, RSV-infected group. The goblet cell analysis at day 6 post-inoculation showed that, around the larger cartilaginous airways, there was a trend towards reduced GC numbers that was not observed in the smaller non-cartilaginous airways. Together with our earlier report that showed good lung deposition of aerosolised palivizumab in lambs [12], this proof-of-principle study provides support for the potential feasibility of an aerosolised therapy approach to treat RSV infection.

While the lambs in our study did not exhibit severe RSV clinical symptoms, as is consistent with previous studies, we found significant differences in the inflammation/congestion scoring when comparing the treated and untreated lambs infected with RSV. In studies by Derscheid and Ackermann and Larios Mora et al., moderate-to-severe necrotic bronchitis, bronchiolitis, and syncytial formation were reported for both the RSV A2 and M37 strains [14,15,16]. One explanation for this difference in clinical symptoms is that the newborn lambs in our model were not colostrum-deprived. Colostrum-deprived models would have prevented the transfer of maternal immunoglobulin to the newborn lambs, potentially making them more susceptible to severe RSV disease [13,21]. We housed the newborn lambs with the ewes for the full study duration to achieve similarity to the human context.

Increased mucus production within the airway is also a known consequence of lung inflammation due to RSV infection. Mucin-producing cells or goblet cells are highly responsive in the respiratory tract, with rapid hyperplasia/metaplasia and hypersecretion in response to inflammatory mediators [22,23]. Given the limited knowledge on mucin production in RSV lamb models, our study analysed goblet cell counts with two approaches: comparing the cell counts in deeper non-cartilaginous bronchioles and in upper airway cartilaginous bronchi to determine the effectiveness of aerosolised palivizumab in terms of reducing inflammation. Although not statistically significant, lower GC counts in the treated lambs compared to the untreated lambs were observed in the cartilaginous airways, but not in the non-cartilaginous airways. The cartilage seen within the lungs was restricted to the upper airways, and the relevance of palivizumab delivery in the upper airway vs. the lower airway is still unknown. Any effect of palivizumab is likely to be within the lower airways, where RSV bronchiolitis occurs. Therefore, further studies are required in order to investigate this effect.

The aerosolised delivery route is a promising approach, as it is non-invasive and directly targets the lung, providing a way to maximise the exposure of pathogens to therapeutic agents. ALX-0171, a trivalent nanobody developed for RSV treatment, was shown to have an anti-viral effect on RSV lung lesions accompanied by a reduction in disease symptoms [21]. Recently, aerosolised delivery methods have been explored for SARS-CoV-2 to block transmission by targeting the upper respiratory tract [24]. We previously demonstrated the pulmonary delivery of aerosolised palivizumab using a lamb model, with over 88% delivery of this mAb to all regions of the lungs, suggesting that the aerosolisation of palivizumab is likely to have an effect in the lower airways [12]. Therefore, aerosolised delivery of palivizumab may be effective for treating RSV infection as well as reducing the currently required dosage, which would reduce cost and thereby improve access in low-resource settings. Further dose-sparing studies are needed in order to address this important question, as well as the potential for this approach to be applied to the newer long-acting monoclonal antibody Nirsevimab.

This study has some limitations. The sample size, while similar to other lamb studies, was relatively small and may have prevented us from detecting any beneficial effects [14]. Our lack of viral load detection in the BALF specimens could be due to variability in the collection method or to not being able to collect across all lung regions, potentially missing areas of RSV infection. The optimal therapeutic dose was not determined in this study and requires further investigation. Despite these limitations, the therapeutic delivery of palivizumab may still be a useful approach to limit the severity of RSV infection. The long-acting version of palivizumab was not available when our study was designed, but could also be considered in future studies. Future studies comparing aerosolized vs. intramuscular palivizumab would also be useful. The advantage of this mode of therapeutic administration is that it can be given at hospital admission even if it has been several days after disease onset. This may allow for its more widespread use in settings where it is difficult to predict the RSV season or where the long-acting mAb is unlikely to be used, at a dose and cost that is more equitable for many LMICs.

## 5. Conclusions

This study has contributed to our understanding of the effect of aerosolised delivery of palivizumab as a therapeutic agent in a neonatal lamb model of RSV infection. Our model did not present any altered clinical symptoms of human RSV disease, but recapitulated features of RSV infection, including lung inflammation. Our aerosolised delivery of palivizumab was able to reduce RSV inflammation, suggesting that the aerosolised delivery of RSV antibody as a mode of therapy is a promising approach that warrants further exploration.

## Figures and Tables

**Figure 1 viruses-15-02276-f001:**
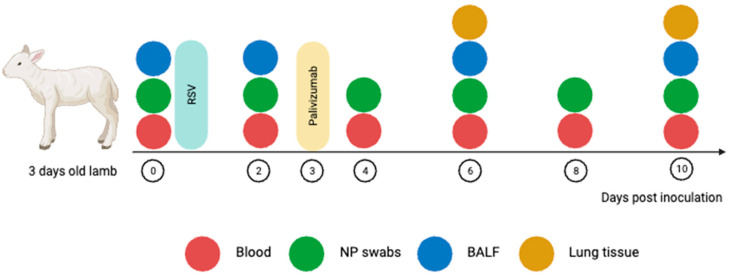
Study design and sampling timeline using an RSV lamb model. Neonatal lambs were infected with the RSV-A2 strain intranasally on day 0. Nasopharyngeal (NP) swabs, blood, and bronchoalveolar lavage fluid (BALF) were collected at baseline (day 0) and then on day 2, 4, 6, 8 and 10 post-inoculation. Lambs were euthanised either on day 6 (peak infection) or day 10 (recovery) post-inoculation according to the study group (refer to Table 1). Image was created using BioRender.com.

**Figure 2 viruses-15-02276-f002:**
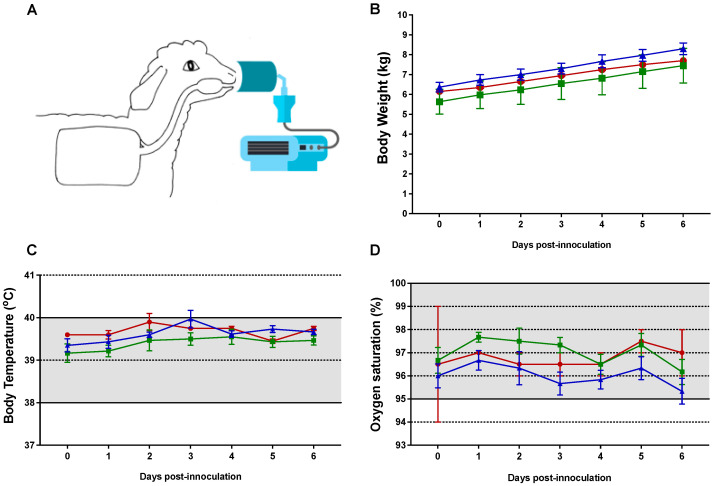
Clinical parameters of newborn lambs given aerosol treatment with palivizumab. (**A**) Nebuliser setup with face mask; (**B**) body weight; (**C**) body temperature; and (**D**) oxygen saturation for each group of lambs. The normal physiological range is shaded in (**C**,**D**). Red circles = control group 0 (*n* = 2); green squares = RSV/untreated group 1 (*n* = 6); blue triangles = RSV/palivizumab group 2 (*n* = 6). Data are presented as the mean ± SEM.

**Figure 3 viruses-15-02276-f003:**
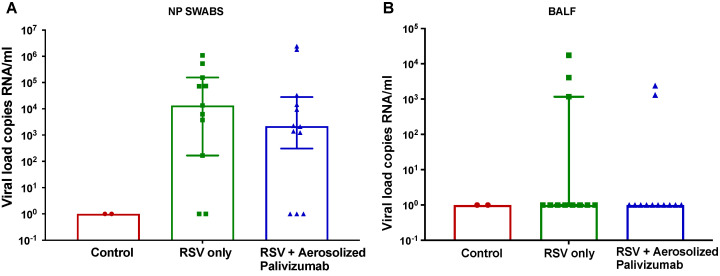
Comparison of RSV load in NP swabs and BALF samples on day 6 post-inoculation (time of peak-infection). (**A**) Nasopharyngeal (NP) and (**B**) bronchoalveolar lavage fluid (BALF) viral load are displayed as averages of all lambs in each group. Red circles = control group 0 (*n* = 2); green squares = RSV/untreated group 1 (*n* = 12); blue triangles = RSV/palivizumab group 2 (*n* = 11). Data are represented as the median ± IQR.

**Figure 4 viruses-15-02276-f004:**
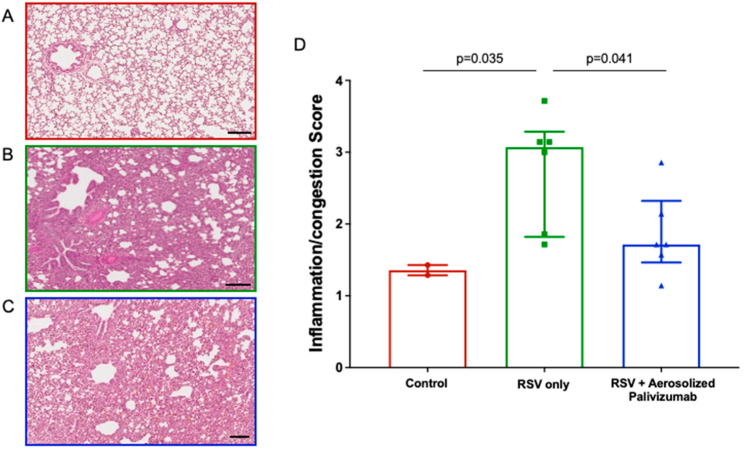
Comparison of inflammation/congestion scores in lambs. (**A**–**C**) Representative images (H&E stain, day 6 post-inoculation) in (**A**) control group lambs; (**B**) RSV/untreated group 1a lambs; and (**C**) RSV/palivizumab group 2a lambs. Scale bar = 100 μm. (**D**) Analysis of inflammation/congestion scores in each group at day 6 post-inoculation (H&E stain, median ± IQR). Red circles = control group 0 (*n* = 2); green squares = RSV/untreated group 1a (*n* = 6); blue triangles = RSV/palivizumab group 2 (*n* = 6).

**Figure 5 viruses-15-02276-f005:**
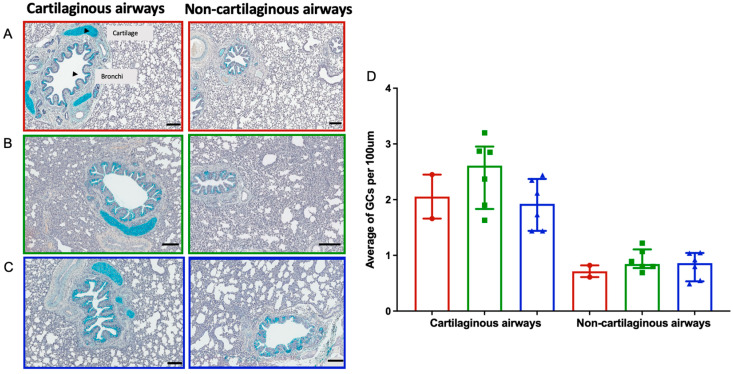
Goblet cell (GC) counts in cartilaginous and non-cartilaginous airways of the lambs. Representative images (AB-PAS stain, day 6 post-inoculation) of cartilaginous airways and non- cartilaginous airways in (**A**) control group 0 lambs; (**B**) RSV/untreated group 1a lambs; and (**C**) RSV/palivizumab group 2a lambs. Scale bar = 100 μm. (**D**) Mean number of GC counts per 100 μm in each lamb from each group at day 6 post-inoculation (AB-Pas stain, median ± IQR). Red circles= control group 0 (*n* = 2); green squares = RSV/untreated group 1a (*n* = 6); blue triangles = RSV/palivizumab group 2a (*n* = 6).

**Table 1 viruses-15-02276-t001:** Study groups.

Group No.	Group Name	RSV Inoculation	Aerosolised Palivizumab	Study Duration (Days)	No. of Lambs
0	Control	-	-	6	2
1a	RSV/untreated–D6	+	-	6	6
1b	RSV/untreated–D10	+	-	10	6
2a	RSV/palivizumab–D6	+	+	6	6
2b	RSV/palivizumab–D10	+	+	10	5

D: day post-inoculation.

## Data Availability

Data is available from the corresponding upon reasonable request.

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
