# Peer review of "Aerosol Delivery of Palivizumab in a Neonatal Lamb Model of Respiratory Syncytial Virus Infection"

_viruses, 2023, doi:10.3390/v15112276_

Round 1
Reviewer 1 Report
Comments and Suggestions for Authors
In this brief report, the authors proposed aerosol delivery of the anti-RSV monoclonal antibody and studied this route in a neonatal lamb model of RSV infection. The paper presents interesting results, and the manuscript is well-written. There are several issues that need to be addressed prior to publication:
1) From my point of view the evaluation of viral loads in collected lung tissue could supplement the experimental data about RSV infection progression.
2) Do the authors plan to include in future experiments the group of lambs that receive the intramuscular dose of palivizumab to compare the beneficial effect of intranasal and systemic introduction routes for palivizumab?
3) Which immune mechanisms may be involved to combat the viral activity in case of using intranasal inhalation treatment?
4) Why do you choose the dosage 113 (±16) mg/kg of aerosolised palivizumab? And how could the limits be increased?
Reviewer 2 Report
Comments and Suggestions for Authors
Several lines of evidence indicate that reduction of viral load is not sufficient to prevent severe RSV disease, including clinical data on motavizumab, Ablynx ALX-071, lumicitabine, presatovir. The sheep RSV model does not generate a highly inflammatory airway response, but is nonetheless a reasonable model. Inhaled delivery is novel. The results reported here confirm the prior literature: reduction in viral load is not sufficient to eliminate airway inflammation.
Choice of neonate sheep is interesting. Follow up work on the response to infection in older animals (sensitized by neonate infection vs naive) may shed light on mechanisms linking RSV infection to childhood wheezing.
